# Interactive Semantic Interventions for VLMs:
# A Human-in-the-Loop Investigation of VLM Failure

**Lukas Klein**[* 1,2,3]**, Kenza Amara**[* 1,4]**, Carsten Lüth**[*2,3,5]**,**
**Hendrik Strobelt** [6]**, Mennatallah El-Assady**[1]**, Paul Jäger**[2,3]

[1] ETH Zürich, Department of Computer Science, Switzerland
[2] German Cancer Research Center (DKFZ), Interactive Machine Learning Group, Germany
[3] Helmholtz Imaging, DKFZ, Germany
[4] ETH AI Center, Switzerland
[5] Heidelberg University, Department of Computer Science, Germany
[6] IBM Research, USA

lukas.klein@dkfz.de, kenza.amara@ai.ethz.ch, carsten.lueth@dkfz.de

## Abstract

Vision Language Models (VLMs), like ChatGPT-o and LLaVA, exhibit exceptional versatility across a wide array of tasks with minimal adaptation due to their ability to seamlessly integrate visual and textual data. However, model failure remains a crucial problem in VLMs, particularly when they produce incorrect outputs such as hallucinations or confabulations. These failures can be detected and analyzed by leveraging model interpretability methods and controlling input semantics, providing valuable insights into how different modalities influence model behavior and guiding improvements in model architecture for greater accuracy and robustness. To address this challenge, we introduce Interactive Semantic Interventions (ISI), a tool designed to enable researchers and VLM users to investigate how these models respond to semantic changes and interventions across image and text modalities, with a focus on identifying potential model failures in the context of Visual Question Answering (VQA). Specifically, it offers an interface and pipeline for semantically meaningful interventions on both image and text, while quantitatively evaluating the generated output in terms of modality importance and model uncertainty. Alongside the tool we publish a specifically tailored VQA dataset including predefined presets for semantic meaningful interventions on image and text modalities. ISI empowers researchers and users to gain deeper insights into VLM behavior, facilitating more effective troubleshooting to prevent and understand model failures. It also establishes a well-evaluated foundation before conducting large-scale VLM experiments. The tool and dataset are hosted at: https://github.com/IML-DKFZ/isi-vlm.

## 1 Introduction

Vision Language Models (VLMs) such as LLaVA [13], LLava-1.5 [11], GPT4-Vision [15] and PaliGemma [4] enable to use the text interface of Large Language Models (LLMs) such as LLAMA [23, 22, 6], GPT4 [15] and Gemini [21] with the visual understanding of CLIP-style image encoders [17] for vision tasks as multi-modal generalist models. The generative language nature of their outputs

---

[*]Contributed Equally

Preprint. Under review.

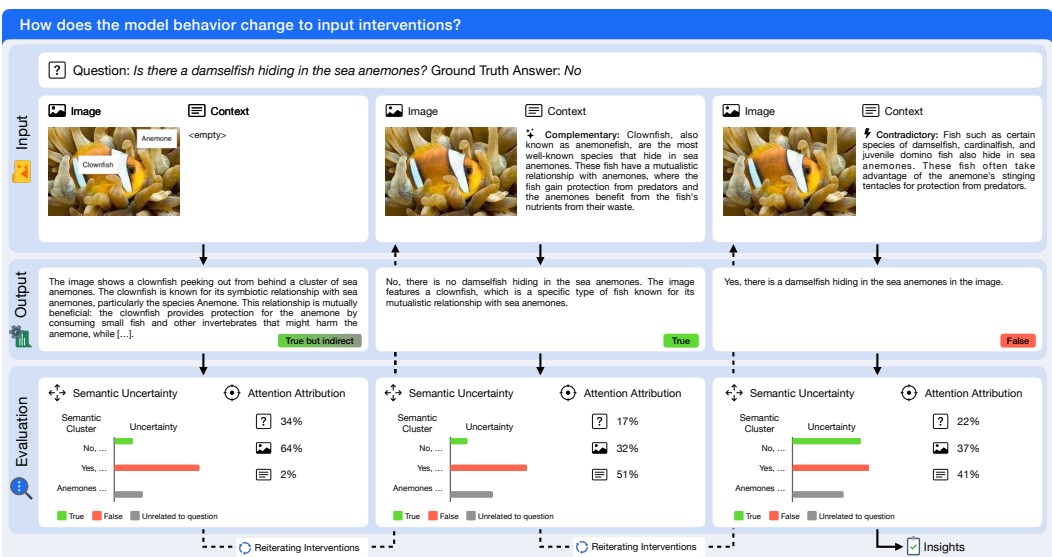

Figure 1: Exemplary intervention workflow with ISI for VLMs in which we iteratively define the input, observe the evaluation results and adapt our intervention until the VLM fails. We start with an annotated image and no context, resulting in a very indirect answer with high attention to the image. We remove the image annotations and add complementary context, resulting in a low uncertainty true answer with larger attention to the context than to the image. By switching to a contradictory context the model fails due to a confabulation provoked by the added context and has high uncertainty for all sampled semantic clusters (clusters are only examples). This exemplary workflow shows how hypotheses about model behavior can be supported by iteratively adapting the inputs. For a more detailed demonstration, please refer to our video available here: `https://drive.google.com/file/d/1LGZBGOG1wK2-qqdSzqD-esvsEk8qVn_v/preview`.

makes them well suited for a wide variety of use cases and their multi-modal natures make them especially uniquely suited to use additional context information from both text and image modality to solve diverse tasks such as content summarizing [14], question answering [26] or text-guided object detection [5]. Especially in the context of Visual Question Answering (VQA), VLMs have demonstrated their capabilities across various applications, such as enabling radiologists to generate medical reports from CT scans [25, 19].

However, current VLM models continue to struggle with the issue of "hallucinations", where the content generated is not faithfully aligned with the provided source material. This widespread problem significantly hinders the large-scale adoption of VLMs in tasks such as VQA. For example, VLMs are like LLMs prone to 'confabulations' [3], a subgroup of "hallucinations", where the answer is sensitive to task-irrelevant information in the text or image modality [8]. To tackle the problem of hallucinations such as confabulations, we have to understand the cause of such model failures. Current research remains limited in understanding how VLMs process information from different modalities, respond to changes in semantics, and how these factors influence their reasoning and answer generation processes. Such understanding is not only essential for diminishing the risk of model failure and increasing the reliability and accuracy of the model but also allows for more effective integration of data modalities in future model architecture developments.

To this end, we developed a tool for "Interactive Semantic Interventions" (ISI), which is accompanied by a VQA dataset including 100 observations with semantic intervention presets. It is designed for interactive evaluation of how interventions on the image and text inputs affect the generated output. We organize the input into three components—image, language context, and question—to analyze the impact of semantic changes within each input. To further quantify the effect of interventions, we compute for each output the model uncertainty via the Semantic Entropy [8] and semantic clusters, as well as the modality importance via the attention attributed to each input [17]. Figure 1 illustrates an example of how interventions can be applied to different inputs, enabling the measurement of their effects and then iteratively adapting these interventions to test specific hypotheses about the model's

behavior. The primary goal is to facilitate rapid, small-scale experimentation, allowing researchers to develop an intuitive understanding of VLM behavior under semantic interventions. Gaining these insights on a smaller scale is essential before undertaking large-scale studies or project development, as it helps in identifying the most effective approaches for probing these models.

Our main contributions are therefore twofold:

**ISI for VLMs**, an interactive tool allowing for diverse interventions on each input, measuring the effect on the output in terms of model uncertainty and input importance. Users can either upload their images and questions or select one of the 100 provided observations, including semantic intervention presets, from our ISI VQA dataset.

**ISI VQA dataset**, including 100 observations with handcrafted intervention presets on the input image and context. Each observation includes an image and annotated image, complementary and contradictory context, and a question with a ground truth Yes/No answer, only answerable through the image.

## 2 Related Work

**VLMs** Recent advancements in VLMs focus on enhancing the integration of visual and textual data, leveraging large multimodal datasets to improve generalization. Current vision-language model architectures, like LLaVA [13], GPT4-Vision [15], and PaliGemma [4], focus on deeply integrating image and text processing through interleaved or token-based strategies within transformer frameworks to enhance multimodal understanding. For example, LLaVA utilizes a simple yet effective strategy by adding vision embeddings as tokens to a pre-existing language model, which allows it to process images as if they were part of the language input, making the model more flexible and reducing the complexity of alignment between modalities. LLaVA-NeXT [12] extends LLaVA by improving image resolution handling, integrating stronger language models, and using more diverse, high-quality training data to enhance its performance in multimodal tasks.

**Interactive Tools for VLMs** Current interactive tools for VLMs primarily focus on facilitating interactive output generation and advancing the field of explainable AI, enabling users to engage dynamically with model outputs. For interactive using generative models, tools like Gradio [1] or Dash [10] allow to quickly build web applications to interactively experiment with VLM models in real-time. More advanced tools are increasingly focusing on explaining VLM outputs by leveraging techniques such as saliency maps, which highlight attention to specific tokens [20], or clustering of hidden states, providing deeper insights into how the model processes and interprets input data [2]. However, current tools offer only limited capabilities for interventions, particularly on image data, and lack mechanisms for semantic-level interventions. Moreover, evaluations predominantly focus on local attention visualizations related to tokens, neglecting broader explanations such as the overall importance of each modality or safety-critical aspects like model uncertainty.

Figure 2: Exemplary observation of the proposed ISI VQA dataset with complementary and contradictory context as well as annotated image presets. This allows direct observation of the behavior of VLMs when given supporting or contradictory information.

## 3 Implementation

We provide a comprehensive summary of the primary methods utilized within the tool and present an overview of the collection and compilation processes involved in the creation of the ISI VQA dataset.

**Vision-Language Models** Per default, our tool incorporates the three VLM architectures LLaVA 1.5, LLaVA-Vicuna, and LLaVA-NeXT, each loaded

from Hugging Face [24]. LLaVA-Vicuna is a version of LLaVA 1.5 leveraging the Vicuna LLM, which is a conversation-fine-tuned version of LLaMA. LLaVA-Vicuna and LLaVA-NeXT both utilize dynamic high resolution for the image input [11], increasing visual reasoning and optical character recognition (OCR) capabilities. Each architecture can be loaded with 7B, 13B, or 32B parameters and 4bit quantized, reducing VRAM and increasing efficiency. For all models, we use conditional probability distribution sampling for the next token prediction due to VRAM and efficiency reasons, but Beam search or Greedy search can be used as well.

**Semantic Entropy**   For quantifying model uncertainty, we employ semantic entropy [7], which calculates entropy based on the sum of token likelihoods between different semantic clusters. For semantic clustering, we use the DeBERTa [9] entailment model. The number of the sampled outputs as well as the sampling temperate $T$ can be selected in the tool, but are by default set to 10 and 0.9. For entropy computation, we use Rao's quadratic entropy [18], which is defined between 0 (no uncertainty) and infinity.

**Attention Attribution**   For attention computation to each input we implemented hooks for the LLaVA architectures to get the start and end positions of the question-, image-, and context-tokens. Due to the dynamic high resolution, the amount of image tokens can drastically differ even for minimal perturbations to the image. Subsequently, we aggregate the attention scores across all attention heads for tokens corresponding to each input and then normalize these values to obtain the relative attention allocated to each input.

**ISI VQA Dataset**   The ISI VQA dataset is a closed-question VQA dataset consisting of 100 observations. Each observation consists of an image, question, and ground truth answer (Yes/No) pair, as well as one text annotated image, one contradictory context, and one complementary context. The context is always in relation to the ground truth answer, aiming to either confuse or help the model without stating an explicit answer to the question, as each question can only be answered through the image. Thus, the model has a very limited ability to leverage prior knowledge to answer the question and must utilize the image input, as prior research indicates a significant bias towards text in VLMs [16]. The annotation on the image gives text-written hints to the ground truth answer. See Figure 2 and Appendix A for exemplary observations. The images are open-source and from the MMMU Benchmark [26]. All other modalities are carefully crafted for the dataset, focusing on quality not quantity.

## 4   ISI for VLMS: A visual analysis workspace

The interactive tool can be used to analyze VLMs with the provided ISI VQA dataset and follows a main pipeline that consists of three main steps: 1) Data & Model Selection 2) Interventions on Image, Context, and Question, and 3) Evaluation. Figure 3 gives an overview of this pipeline. While we showcase each step of the pipeline based on a provided example from the ISI VQA dataset, subsection 4.3 explains how to use the tool with custom data. A detailed look at the User Interface (UI) can be found in Appendix C and a visualization of outputs and perturbations on an example use-case in Appendix D.

### 4.1   General Information

**Users**   The application is catered toward researchers, developers, and other users with a basic understanding of VLMs, who are interested in interpreting model behavior through semantic interventions on VLMs. By enabling fast-paced iterations in a human-in-the-loop scenario, it allows the building of intuitions before scaling experiments in large-scale projects.

**System Requirements**   ISI for VLMS is an interactive tool embedded in a locally hosted web application requiring a computer with sufficient VRAM for VLM inference. The minimum required VRAM for a 4bit-quantized LLAVA 7B model is around 8GB while LLaVA-Vicuna and LLaVA-Next require 12GB. The computation of the semantic entropy with the DeBERTa model requires an additional 7GB. The exact amount of VRAM depends on the amount of input tokens.

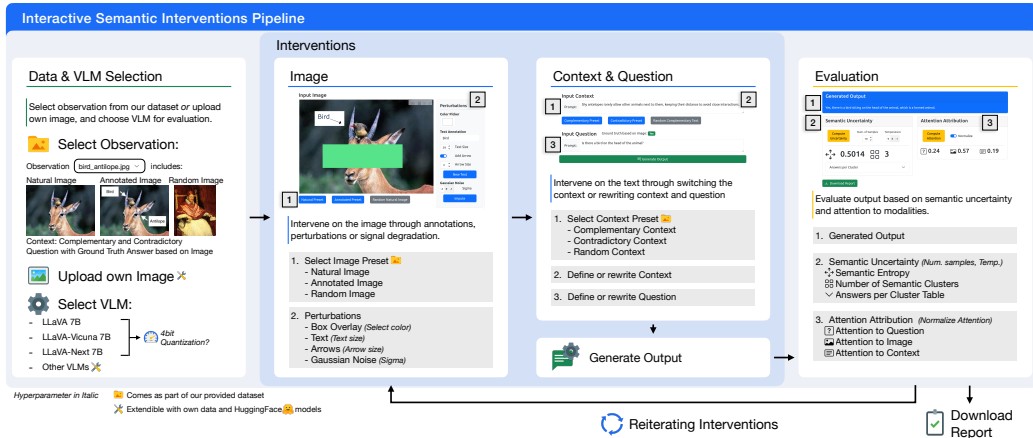

Figure 3: Illustration of the evaluation pipeline used in the ISI for VLMs to enable interactive exploration of VLM behavior under various scenarios. It consists of three main stages: 1) Data & VLM Selection: Users choose an observation either from the provided ISI VQA dataset or upload their own, and select a VLM for evaluation. 2) Interventions on Image, Context & Question: The selected image can be altered through presets or perturbations, and the context or question can be edited or also switched with presets. 3) Evaluation: The output is analyzed for semantic uncertainty and attention attribution, allowing for iterative refinement of interventions. For more information see section 4 and Appendix C for a detailed visualization of the UI.

## 4.2   Interactive Semantic Intervention Pipeline

**Data & Model Selection:**   As the first step, a user either chooses an observation from the ISI VQA dataset or uploads their custom image. Each observation from the dataset includes an image, corresponding context, and a question with a ground truth answer, as well as the presets for the annotated images and contradictory and complementary context. The corresponding image, context, and question are displayed.  In the next step, the user selects a VLM (LLaVA, LLaVA-Vicuna, LLaVA-Next) and the number of parameters (7B, 11B, 32B) in two separate drop-down menus. 4-bit quantization can be enabled to reduce the computational load and VRAM requirements.

**Interventions on the Image:**   For interventions on the image, ISI allows the user two main functionalities. First, on the proposed ISI VQA dataset the user can select for each observation three different image presets (natural image without modifications, annotated image with hand-crafted annotations, and random natural image from the dataset) by selecting the respective buttons. Second, ISI allows perturbing the image directly in the tool by overlaying boxes with selectable colors, inserting and modifying text, adding directional arrows, and introducing Gaussian noise with adjustable noise values.

**Interventions on Context & Question:**   To facilitate the user's ability to observe how various contexts and questions affect the model's performance two functionalities are supported.  In the proposed ISI VQA dataset, users can choose from three distinct context presets—complementary, contradictory, or random—by selecting the respective button, automatically updating the content in the text input fields. Additionally, the user can manually edit the context and question in these fields.

**Evaluation**   The evaluation is designed to enable quantitative analysis of how interventions on image and text impact the behavior of the selected VLM. At the top, the current input is visualized to always relate the evaluation results to the correct input.  Below, the generated output, semantic uncertainty, and attention attribution are shown. For computational reasons, each evaluation can be started separately.

The semantic uncertainty tab allows users to evaluate model uncertainty by clustering sampled outputs according to their semantic meaning. This feature highlights the range of semantic differences in the outputs and calculates semantic entropy, providing a comprehensive view of the model's overall uncertainty. It displays key metrics, such as semantic entropy and the number of semantic clusters,

which are influenced by adaptable hyperparameters like the number of samples and the sampling temperature. For deeper exploration, the "Answers per Cluster" dropdown provides a table displaying all sampled answers along with their assigned semantic clusters. This table enables users to examine the full range of generated outputs and understand the semantic similarities within each cluster. To evaluate the significance of each of the three inputs during generation, the attention attribution tab displays the absolute or relative attention assigned to the question, context, and image input tokens.

To provide a contextual understanding of the current observation, the tool additionally displays average values for attention attribution and semantic entropy across the entire ISI VQA dataset based on each VLM architecture. These averages are shown when hovering over the relevant values.

**Export** The results of one iteration can be exported as a PDF to facilitate the systematic collection of example cases for further analysis and to support the transition from initial qualitative insights to small-scale quantitative evaluation. After the analysis, users can download a comprehensive report that includes the image, context, question, detailed model setup, hyperparameters, and all computed evaluation metrics.

### 4.3 Custom Data & Extension of VLMs

**Custom Observations** Through simple drag and drop or image selection in the file explorer, users can easily use the entire possibilities of the pipeline with the exceptions of the presets provided in our ISI VQA dataset. The corresponding context and question can be added in the text fields. To use ISI with a large dataset, we recommend replacing or extending the CSV file provided with the ISI VQA dataset and adding the images to the 'natural images' folder. This allows the user to select observations via drop-down and instantly load question, context, or image data. See our code repository for a detailed description.

**Extending VLMs** The tool can easily incorporate additional VLMs that are implemented in the Hugging Face framework by adding a new option in the model selection code and the respective drop-down menu. Depending on the VLM architecture, the attention attribution and the prompt template have to be adapted as well. See our code repository for a detailed description.

## 5 Discussion & Conclusion

ISI for VLMs offers a practical, hands-on approach to investigate when VLMs generate false statements and hallucinate under input perturbations by reintroducing human input into the experimental cycle. By allowing users to implement controlled interventions on both image and text inputs, the tool facilitates rapid, small-scale, and iterative experimentation. This approach is essential for developing and refining intuitions and hypotheses about model behavior under various conditions.

These interpretability features are particularly valuable for identifying scenarios in which VLMs may fail, e.g. when context and image information are in conflict. Understanding model failures is crucial, especially in cases of hallucinations and confabulations, where models produce outputs that do not accurately reflect the input data. The ability to fully control input semantics and evaluate model uncertainty allows for the detection and deeper understanding of such hallucinations. Beyond identifying model failures, ISI can be utilized to analyze how different modalities influence model behavior and performance, enabling the adaptation of model architectures, data sampling, and transformations to increase accuracy and robustness. Overall, ISI serves as a strong foundation not only for planning large-scale studies but also for gaining initial insights into the behavior of different VLM models from an educational perspective.

The main limitation of ISI is that for serving the VLM a GPU is required which is costly and why we are not providing it in the form of an openly accessible website.

In summary, ISI empowers researchers and end-users to gain deeper insights into VLM behavior and their failures, enabling more effective troubleshooting and refinement of these models. By facilitating intuitive, iterative experimentation, ISI allows researchers to advance the reliability and robustness of VLMs in production.

## Acknowledgments

This work was funded by Helmholtz Imaging (HI), a platform of the Helmholtz Incubator on Information and Data Science.

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

# Appendix

## A ISI VQA Dataset

The dataset is a hand-crafted closed-ended [yes/no] Visual Question Answering (VQA) dataset with 100 observations. Each question is designed to only be answered clearly with the information provided in the image. Each image-question pair gets two additional text context descriptions which either give general information that is supposed to 'nudge' the model to the correct (complimentary) or incorrect (contradictory) answer. In addition, we provide a text annotated preset of the image which annotates objects in the image, making answering the question easier. All images are open-source from the MMMU Benchmark [26].

A list of examples is visualized in Figure 4.

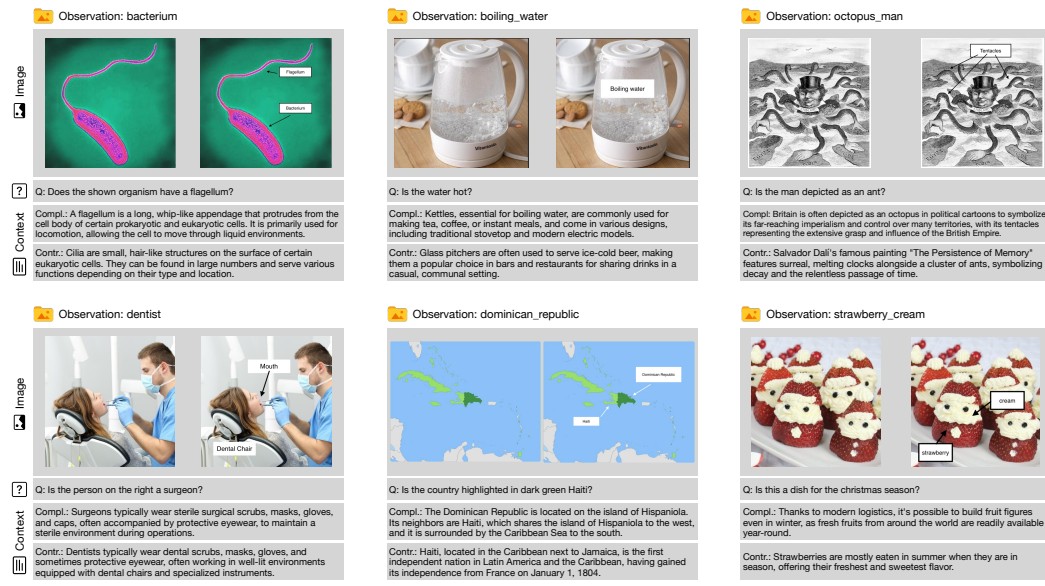

Figure 4: Exemplary overview of observations in the contributed VQA dataset.

# B  Setup

The code is located at: `https://gitlab.com/dekfsx1/isi-vlm`

We recommend Conda for handling the environments.

```
# Create environment
$ conda create -n isi-vlm python=3.10
$ conda activate isi-vlm

# Download
$ git clone https://gitlab.com/dekfsx1/isi-vlm
$ cd isi-vlm

#Setup
$ pip install torch==2.3.0 torchvision==0.18.0 torchaudio==2.3.0 \
--index-url https://download.pytorch.org/whl/cu118
$ pip install .

#Execution
$ python dash_app/app.py
```

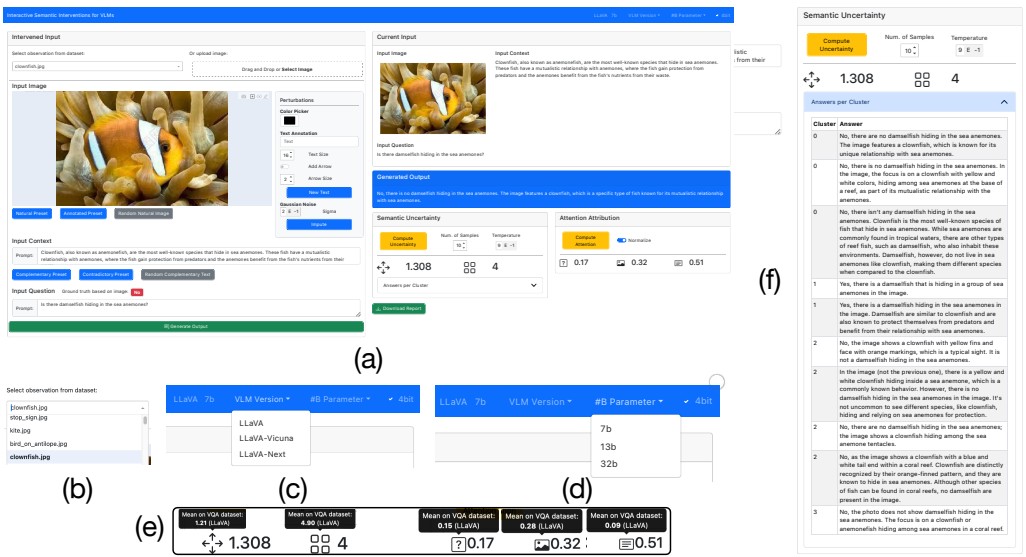

Figure 5: Overview of the UI functionality. A screenshot of all parts of the tool with an exemplary input is shown in (a). The drop-down menu with text search is shown in (b) and the drop-down VLM selection for family and number of parameters is shown in (c) and (d). The hover windows with mean aggreagted values for the semantic entropy, semantic clusters and attention attribution is shown in (e). Exemplary outputs of the answers per cluster visualization along with a more detailed view of the part can be seen in (f).

## C  Interface

The user interface is shown with explained functionality in Figure 5.

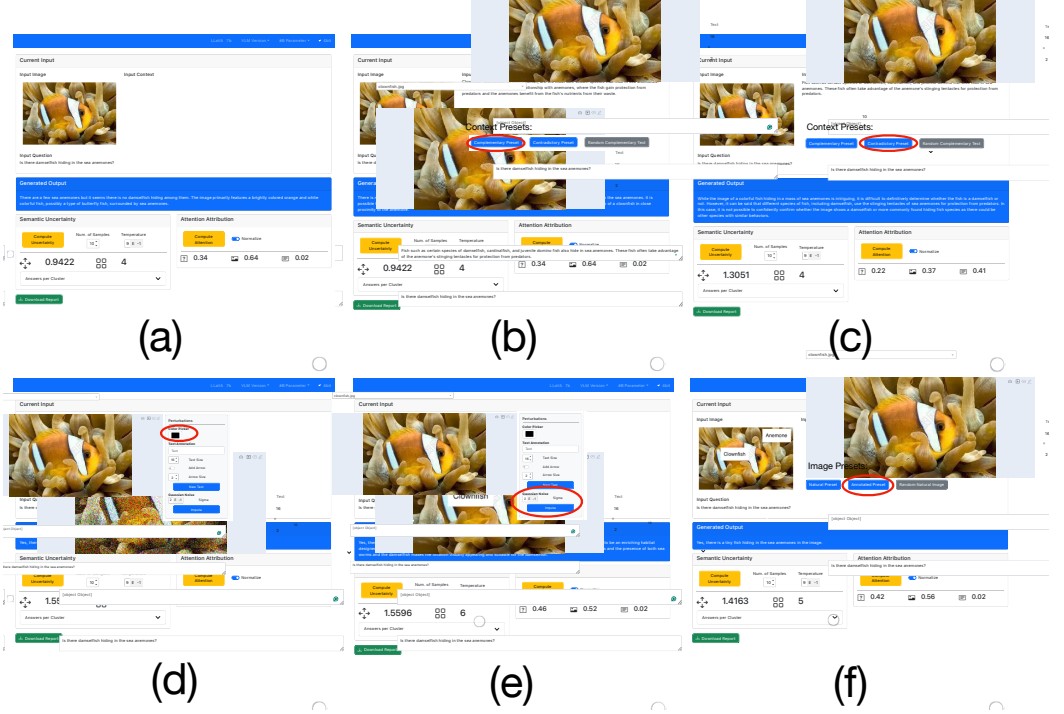

Figure 6: Example of an investigation using ISI for VLMs showing the difference in model behavior for an empty context (a), the same image with the complimentary context preset (b) and contradictory context preset(c). The output of the model for the image with an additional blox box is shown in (d) and in (e) for the image with gaussian noise distribution while for (f) the annotated preset with text box annotations of important objects.

# D Example Investigation

An example investigation of single case is shown in Figure 6.

