# OpenReview forum: "Interactive Semantic Interventions for VLMs: A Human-in-the-Loop Investigation of VLM Failure"
_NeurIPS.cc/2024/Workshop/SafeGenAi — SafeGenAi Poster_

### Official Review · Reviewer_cZKt · 2024-10-08
**A review on Interactive Semantic Interventions for VLMs: A Human-in-the-Loop Investigation of VLM Failure**

**Rating:** 6
**Confidence:** 4

**Review:**

Pros: The tool presents a user-friendly interface with comprehensive functionality, affording users a high degree of freedom for interactive and iterative testing of Visual Language Models (VLMs). This flexibility is beneficial for researchers and developers seeking to explore model behaviors in-depth. Furthermore, the inclusion of a report export feature adds significant value by facilitating the systematic collection and analysis of test results, thus supporting a streamlined transition from initial insights to quantitative evaluation.

Cons: The tool's current reliance on manual interaction may limit its efficiency, as it requires continual human operation to identify issues. Enhancing automation through the integration of components like LLM-as-a-Judge could expedite problem detection. Additionally, the tool's current configuration only supports local inference, which restricts access for users without high-performance hardware.

---

### Official Review · Reviewer_pkkT · 2024-10-08
**A review of Interactive Semantic Interventions for VLMs: A Human-in-the-Loop Investigation of VLM Failure**

**Rating:** 6
**Confidence:** 3

**Review:**

Summary: This paper introduces a tool called Interactive Semantic Intervention (ISI), which aims to help researchers understand and improve the performance of VLMs in multimodal tasks, especially the "hallucination" and other erroneous output problems that appear in VQA tasks. By allowing users to perform semantic interventions on image and text inputs, the ISI tool can detect the model's response to input changes and quantify the uncertainty of the output. This method provides new ideas for understanding the process of multimodal information integration and provides a basis for improving model structure and enhancing robustness.

**Strengths**:

1. The article introduces the **ISI** tool, which offers researchers a novel method to observe the behavior of VLMs by adjusting image and text inputs.
2. The article incorporates methods like Semantic Entropy to quantify the uncertainty in VLM outputs. These metrics allow users to more clearly **assess** the decision-making process of the model and its sensitivity to different inputs.

**Weaknesses**:

1. The article sets the VQA task as a closed-ended problem. However, for open-ended, more complex question-answering scenarios, the applicability of the ISI tool may be limited, potentially failing to comprehensively test VLM performance across a wide range of **diverse tasks**.
2. The **interference methods** provided by ISI are relatively limited, focusing mainly on image and text disturbances. If these disturbances could be made dynamic, it might offer researchers a richer perspective on understanding VLM performance across tasks.

---

### Official Review · Reviewer_FjCZ · 2024-10-09
**Review for Interactive Semantic Interventions for VLMs: A Human-in-the-Loop Investigation of VLM Failure**

**Rating:** 5
**Confidence:** 3

**Review:**

Strength: Well designed tool with very interesting objective.

1. The paper introduces a valuable tool and dataset for explainable interpretation of Vision-Language Model (VLM) behavior in response to various perturbations. Users can interact with the system by applying image perturbations (such as bounding boxes or noise) and modifying the textual context of questions (either complementary or misleading) to observe changes in semantic uncertainty or attention attribution across different modalities.
2. The tool is well-designed, supporting multiple VLMs, featuring an intuitive user interface, and offering customizable options for enhanced user interaction.

Weakness: Lack of Practical Application

While the concept of interactive semantic interventions (ISI) is intriguing, the paper does not effectively demonstrate how the results of the analysis (such as attention attribution and uncertainty measures) can be applied to solve real-world issues in Vision Language Models (VLMs). The fine-grained insights provided by the tool may seem insightful, but there is no clear demonstration of how these insights translate into practical improvements in VLM behavior.

The example provided in figure 1 (clownfish hiding in sea anemones) seems to present an intuitive scenario where complementary text would naturally improve the model's accuracy. The change in attention to different modalities seem intriguing: unfortunately, I can't imagine how I could practically use such implication.

The paper would benefit from a simple experiment showing how the tool can be used to improve VLMs or guide their fine-tuning. Even a very simple scenario would be nice to help the reader imagine the potential of the tool.